# Role of Ganetespib, an HSP90 Inhibitor, in Cancer Therapy: From Molecular Mechanisms to Clinical Practice

**DOI:** 10.3390/ijms24055014

**Published:** 2023-03-06

**Authors:** Mahmoud E. Youssef, Simona Cavalu, Alexandru Madalin Hasan, Galal Yahya, Marwa A. Abd-Eldayem, Sameh Saber

**Affiliations:** 1Department of Pharmacology, Faculty of Pharmacy, Delta University for Science and Technology, Gamasa 11152, Egypt; 2Faculty of Medicine and Pharmacy, University of Oradea, P-ta 1 Decembrie 10, 410087 Oradea, Romania; 3Department of Microbiology and Immunology, Faculty of Pharmacy, Zagazig University, Al Sharqia 44519, Egypt; 4Department of Pharmacology and Biochemistry, Faculty of Pharmacy, Horus University, New Damietta 34518, Egypt

**Keywords:** ganetespib, HSP90, cancer, EGFR, IGF-1, VEGF

## Abstract

Heat-shock proteins are upregulated in cancer and protect several client proteins from degradation. Therefore, they contribute to tumorigenesis and cancer metastasis by reducing apoptosis and enhancing cell survival and proliferation. These client proteins include the estrogen receptor (ER), epidermal growth factor receptor (EGFR), insulin-like growth factor-1 receptor (IGF-1R), human epidermal growth factor receptor 2 (HER-2), and cytokine receptors. The diminution of the degradation of these client proteins activates different signaling pathways, such as the PI3K/Akt/NF-κB, Raf/MEK/ERK, and JAK/STAT3 pathways. These pathways contribute to hallmarks of cancer, such as self-sufficiency in growth signaling, an insensitivity to anti-growth signals, the evasion of apoptosis, persistent angiogenesis, tissue invasion and metastasis, and an unbounded capacity for replication. However, the inhibition of HSP90 activity by ganetespib is believed to be a promising strategy in the treatment of cancer because of its low adverse effects compared to other HSP90 inhibitors. Ganetespib is a potential cancer therapy that has shown promise in preclinical tests against various cancers, including lung cancer, prostate cancer, and leukemia. It has also shown strong activity toward breast cancer, non-small cell lung cancer, gastric cancer, and acute myeloid leukemia. Ganetespib has been found to cause apoptosis and growth arrest in these cancer cells, and it is being tested in phase II clinical trials as a first-line therapy for metastatic breast cancer. In this review, we will highlight the mechanism of action of ganetespib and its role in treating cancer based on recent studies.

## 1. Introduction

When exposed to environmental stressors, most cells produce a small subset of proteins collectively referred to as “heat-shock proteins” or “stress proteins” in considerably increased amounts [1,2,3]. The majority of heat-shock proteins (HSPs) are molecular chaperones that constitutively regulate many of the major regulators of cell growth and survival through affecting their proper folding, intracellular distribution, and proteolytic turnover [4,5,6,7]. When faced with protein-denaturing stressors such as temperature changes, their intracellular expression rises as an evolutionarily conserved response to reestablish the ideal environment for protein folding and to improve cell survival. To maintain homeostasis in a hostile environment, malignant cells surely increase their expression of chaperone proteins, as seen in many different forms of tumors. However, other data suggest that chaperone proteins also enable tumor cells to resist fatal cellular changes within their demanding microenvironments [1,8].

As our understanding of the functions that chaperone proteins play in establishing and maintaining altered phenotypes expands, our interest in pharmacologically modifying chaperone activity to treat cancer and other disorders has also increased [9]. Over the last decade, several small-molecule drugs targeting the molecular chaperone HSP90 have shown potential as anticancer treatments. These drugs have the unique ability to disrupt the function of various receptors, kinases, and transcription factors that are known to play a role in oncogenesis [1].

## 2. Types of Heat-Shock Proteins

According to the classification proposed by Kampinga et al., the human genome encodes a wide variety of HSPs. These include the HSPA (HSP70) family, the HSPH (HSP110) family, the DNAJ (HSP40) family, the HSPB (small HSP) family, the HSPC (HSP90) family, and the chaperonin families (HSP60 and CCT) [10]. For example, the HSPA family has 13 members, the DNAJ family has 40 members, and the HSPB family has 11 members. Additionally, some members have tissue-specific expression, such as HSPA2, which is highly expressed in the testes, while others are housekeeping proteins, such as HSPA8, which is required for co-translational folding and protein translocation across intracellular membranes. In addition to the HSP families, there are other heat-inducible proteins, such as HSP47, that have chaperone-like functions, but are not part of any of the HSP families (Table 1).

## 3. HSP90 Family

HSP90 is a highly conserved protein that is present in all eukaryotic organisms, from yeast to humans. It is one of the largest molecular chaperones and has a molecular weight of around 90 kDa. It is highly expressed in cells that are undergoing rapid growth and division, such as cancer cells, and is required for the stability and activity of many key signaling proteins, including kinases and transcription factors [18]. HSP90 is an ATP-dependent chaperone, meaning that it requires ATP to perform its functions. It binds to its client proteins in an ATP-dependent manner, preventing them from aggregating and promoting their folding into their active conformations [19]. HSP90 can also interact with other molecular chaperones, such as HSP70 and HSP40, to form a complex network of proteins that regulate the folding, maturation and degradation of cellular proteins [20]. In addition to its role in protein folding, HSP90 has been implicated in various cellular processes, including the regulation of cell cycle progression, DNA damage responses, and cell survival. HSP90 also plays a crucial role in the regulation of the activity of signaling pathways that are associated with cancer, such as the protein kinase B (Akt)/mammalian target of the rapamycin (mTOR) pathway, and the rapidly accelerated fibrosarcoma (Raf)/mitogen-activated protein kinase (MAPK) pathway [21].

The HSP90 family is composed of several different members (Table 1), including HSPC1 (HSP90N), HSPC2 (HSP90α), and HSPC3 (HSP90β and HSP90B), the latter of which are the two most well-studied and highly conserved members across species [22]. They are similar in structure and function, but have different patterns of tissue expression. HSPC4 (Gpr94) is another member of the HSP90 family that is found in the endoplasmic reticulum and is involved in the folding and maturation of secreted and transmembrane proteins [23]. HSPC5 (TPR-containing HSP90 tumor necrosis factor receptor-associated protein 1, “TRAP1”) is localized to the mitochondria and has roles in regulating oxidative stress, mitochondrial functions, and cellular metabolism [24]. Cpr6 is a fungal-specific member of the HSP90 family that is involved in regulating cell growth and division [25]. These different members of the HSP90 family share a conserved core of ATPase and substrate-binding domains, but they also have unique accessory domains that allow for the regulation of their activity and specificity, making them a diverse and versatile group of proteins in terms of their functions in the cell.

## 4. HSP90 Structure

In eukaryotic cells, heat-shock protein 90 (HSP90) is one of the most prevalent and conserved molecular chaperones. Unlike other well-known molecular chaperones such as HSP60 and HSP70, HSP90 assists in the final maturation of certain types of proteins, but it is not essential for the de novo folding of most proteins. HSP90’s clients include protein kinases, transcription factors such as p53, and steroid hormone receptors (SHRs). Consequently, HSP90 is involved in several cellular activities, including signal transduction, intracellular transport, protein degradation, and the regulation of protein folding [26,27,28].

HSP90 chaperones are homodimers consisting of three conserved domains. The ATP binding site is located in the N-terminal domain of HSP90-N (NTD), which is involved in the demands of the middle domain’s catalytic loop and the momentary dimerization of the N-terminal domain. The middle domain (HSP90-M, MD) helps in binding with p23 and Aha1 and is necessary for the HSP90-N-mediated activation of ATP hydrolysis. The dimerization domain (HSP90-C, CTD) is located at the C-terminus. It has been proposed that it might help with substrate binding [26,28].

## 5. Mechanisms Regulating HSP90 Machinery

HSP90 machinery can be regulated in two different ways:

ATPase activity: The ATPase activity of HSP90 and cycling between the closed and open states are the cornerstones of its mechanism of action. All HSP90 isoforms, including the homologs in the cytoplasm, endoplasmic reticulum (ER), and mitochondria, act similarly in terms of conformational changes after nucleotide binding, despite being situated in various parts of the cell. As previously mentioned, HSP90 is a flexible homodimer whose monomers are made up of three structural domains: NTD, MD, and CTD. ATP binding to its binding cleft in the NTD of HSP90 triggers a cascade of conformational processes. The translocation of a brief N-domain fragment (ATP-lid) over the binding pocket and the subsequent attachment to the matching N-domain of the opposite homodimer, which results in a twisted and compressed dimer, is a fascinating phenomenon (Figure 1). The outcome is that the N- and M-domains move in with a closer proximity, completing the “split ATPase” site. The N-domains of the HSP90 homodimers separate after ATP hydrolysis, releasing ADP and Pi while the HSP90 resumes its initial open conformation [29,30,31].

Chaperone cycle: The HSP90 chaperone cycle is a convoluted process by which members of the HSP90 family carry out their duty of folding client proteins. The HSP90 chaperone machinery involves the assistance of other molecules, including co-chaperones, partner proteins, and immunophilins, which function in a precise and dynamic manner, aiding the efficient protein folding by HSP90 [29,30].

## 6. Role of Extracellular HSP90 in Cancer

Extracellular HSP90 (eHSP90) is a form of HSP90 that is found outside of cells, and it has been implicated in the pathogenesis of cancer. eHSP90 plays a crucial role in the survival and proliferation of cancer cells by maintaining the stability and function of its client proteins, which are involved in several signaling pathways that regulate cell growth, survival, and angiogenesis [32]. The mechanism by which HSP90 is exported outside of cells is not yet fully understood, although several pathways have been proposed. One possible mechanism involves vesicular trafficking, a process by which proteins are packaged into vesicles and transported to the cell surface for secretion. In this pathway, HSP90 is first synthesized in the cytoplasm and then transported to the endoplasmic reticulum (ER), where it undergoes processing and is packaged into vesicles. The vesicles are then transported to the Golgi apparatus, where they undergo further processing and are ultimately released from the cell through exocytosis [33,34]. Another suggested mechanism for the export of eHSP90 involves direct transport across the plasma membrane. In this pathway, HSP90 is believed to interact with specific membrane transporters or channels that facilitate its export. The precise identity of these transporters or channels is not yet known, but research has suggested that certain HSPs may interact with distinct transporters or channels to promote their export [35].

eHSP90 can exist in multiple forms, including full-length HSP90α (major) and HSP90β (minor) [36], truncated HSP90 due to protein cleavage [37], extracellular vesicles associated with exosomal HSP90 [38], actively secreted HSP90 from cells [39], and surface-bound HSP90 on certain cells [40]. These different types of extracellular HSP90 are involved in intercellular signaling, immune system regulation, and carcinogenic transformation. Secreted HSP90, for example, can trigger certain signaling pathways in target cells, thus being a factor in cancer and autoimmune diseases [35]. HSP90 present in exosomes may act in cell-to-cell communication, while the surface-bound form of HSP90 may play a role in cell adhesion and signaling [41]. All in all, extracellular HSP90 is a complex, dynamic protein with a wide range of biological activities.

eHSP90 has been observed to interact with several receptors and signaling pathways involved in cancer, including the EGFR, vascular endothelial growth factor receptor (VEGFR), and IGF-1R. By binding to these receptors and modulating their activity, eHSP90 can contribute to the survival and growth of cancer cells [42]. Apart from its role in stabilizing client proteins, eHSP90 has also been found to promote angiogenesis, which involves the formation of new blood vessels that are crucial for tumor growth and spread. This is achieved through its interaction with the extracellular matrix and regulation of pro-angiogenic factors such as the vascular endothelial growth factor (VEGF).

Besides its role in promoting tumor cell survival and angiogenesis, eHSP90 has been found to regulate the immune response to cancer. It can interact with immune cells such as dendritic cells and T cells and modulate their activation and function [43]. By regulating the immune response to cancer, eHSP90 can contribute to the evasion of the immune system by tumor cells, promoting tumor growth and progression. Furthermore, several studies have shown that eHSP90 can bind to cancer-associated fibroblasts (CAFs), which are stromal cells that play critical roles in regulating the tumor microenvironment. eHSP90 can modulate the activation and differentiation of CAFs, resulting in changes in the tumor microenvironment that promote tumor cell growth and progression [44]. Additionally, eHSP90 can act as a ligand for some cell surface receptors, such as Toll-like receptors (TLRs) and receptor tyrosine kinases (RTKs), activating intracellular signaling pathways that regulate cellular functions such as cell survival, growth, and differentiation. This interaction with cell surface receptors can play a significant role in regulating the response of immune and stromal cells to tumors, as well as regulating the function of tumor cells themselves [21].

## 7. HSP90 Inhibitors

### 7.1. Natural Products and Their Derivatives

Geldanamycin was the first HSP90 inhibitor discovered in the early 1990s. It works by binding to the N-terminal ATP-binding site, which disrupts the ATPase activity of the molecular chaperone and leads to client protein degradation and interference with multiple cellular processes [1]. Although it is effective against cancer cells, geldanamycin is not a suitable clinical candidate due to toxicity, instability, and poor solubility. Another natural product, radicicol, was isolated from the fungus *Monosporium bonorden* and also binds to the ATP-binding site of HSP90. Despite not being useful in clinical applications, these two natural products demonstrate the potential of HSP90 inhibition as a multi-faceted approach to treating cancer [45].

### 7.2. Semisynthetic HSP90 Inhibitors

In 2003, the first geldanamycin derivative, 17-N-allylamino17-dimethoxygeldanamycin (17-AAG), entered phase I clinical trials, but none of the trials advanced past phase II [46]. Despite the initial promising results against cancer tumor growth, 17-AAG had poor bioavailability [47]. To address this, Infinity Pharmaceuticals designed 17-AAGH2, which improved the metabolic profile and potency of 17-AAG. Despite some clinical trials with 17-AAGH2, half were terminated or withdrawn [48]. Another derivative, 17-dimethylamino-17-dimthoxygeldanamycin (17-DMAG), was developed by Kosan Biosciences to improve the physiochemical profile of geldanamycin [49].

### 7.3. Purine-Based Inhibitors

The first fully synthetic small-molecule HSP90 inhibitor, PU-3, was discovered and was shown to inhibit growth and differentiation in breast cancer cells [50]. Researchers then developed derivatives of the purine chemical scaffold in PU-3, resulting in five purine or purine-like compounds that entered clinical trials, but none advanced past phase II. These inhibitors included BIIB021, BIIB028, MPC-3100, PU-H71, and Debio0932 [51].

### 7.4. Benzamide Inhibitors

The first pyrazole-containing HSP90 inhibitor discovered was SNX-5422. This compound has promising bioavailability as a prodrug, but has the drawback of inhibiting all HSP90 isoforms, leading to dose escalation challenges [52].

### 7.5. Resorcinol-Containing Inhibitors

AUY922 (luminespib), a resorcinol-containing small-molecule HSP90 inhibitor, was first identified by Chueng and colleagues at the Institute for Cancer Research in London. Preclinical studies showed that it was active against tumor growth, angiogenesis, and metastasis in a xenograft mouse model [53]. Synta Pharmaceuticals modified the resorcinol scaffold to create STA-9090, which demonstrated a higher potency in downregulating oncoproteins and pathways compared to first-generation inhibitors. STA-9090 progressed to phase III trials and was the most clinically evaluated HSP90 inhibitor [54]. Another resorcinol-containing HSP90 inhibitor, AT13387, was discovered by Astex Therapeutics in the UK, and optimization led to a resorcinol scaffold with a high potency and ideal ligand efficacy [55]. KW-2478 was discovered by scientists at Kyowa Hakko Kirin in Japan, and ganetespib is another second-generation HSP90 inhibitor composed of the resorcinol moiety of radicicol derivatives [55].

### 7.6. Miscellaneous Inhibitors

Several other research groups have also created inhibitors for the molecular chaperone, but these inhibitors are unique and their structures are not publicly available. These inhibitors include XL888, HSP990, DS2248, and TAS-116. Except for TAS-116, all of these inhibitors work in the same way as the previously mentioned inhibitors, by binding to the NTD of HSP90 [56]. TAS-116 is unique because it was the first reported compound to enter clinical trials that specifically targeted the cytosolic forms of HSP90 (HSP90α and HSP90β). Since the early 2000s, all HSP90 inhibitors that have gone through clinical trials, except for TAS-116, have targeted the NTD of HSP90. Clinical data show that while this approach is effective against tumors, it also has toxic side effects, including liver, heart, and eye toxicity, and limitations in dose scheduling. There is a need to develop new HSP90 inhibitors with an improved effectiveness and a reduced toxicity [57]. All HSP90 inhibitors are summarized in Table 2.

## 8. Ganetespib as HSP90 Inhibitor

Ganetespib is a second-generation HSP90 inhibitor that belongs to the class of radicicol derivatives, which are composed of the resorcinol moiety. Unlike first-generation inhibitors, ganetespib does not contain the benzoquinone moiety, which has been associated with hepatotoxicity. This synthetic small-molecule inhibitor is resorcinol-based and non-geldanamycin, and contains a triazolone moiety (Figure 2) [58].

In vitro, ganetespib demonstrates strong cytotoxicity in several hematological and solid tumor cell lines, including those that express mutant kinases that make a tyrosine kinase inhibitor for tiny molecules resistant. Ganetespib demonstrates continuous activity even after brief exposure times [58,60], in addition to quickly degrading recognized HSP90 client proteins [58]. Ganetespib cytotoxicity in these cell lines is mostly caused by an irreversible commitment to apoptosis, most likely after impacts to the cell cycle and growth arrest [58,60].

In vivo, ganetespib exhibits strong anticancer properties by substantial growth inhibition or regression in hematological tumor models and solid tumor models, respectively. Compared to geldanamycin analogs, ganetespib does not cause the dose-limiting liver damage that has been reported. The ocular toxicity associated with NVP-AUY922 and SNX-522 has not been observed with ganetespib because ganetespib is quickly removed from the retinal tissues and does not build up in the rat eye [58,61,62].

## 9. Molecular Mechanisms of Ganetespib

HSP90-mediated inhibition by ganetespib could lead to antitumor activity via the modulation of various signaling mechanisms. Several oncogenic signal transduction proteins are specified “clients” of HSP90 and substantially rely on its function for maturation and/or stabilization. Notably, HSP90 client proteins include transcription factors, members of the receptor tyrosine kinase family, cell cycle regulators, members of the steroid receptor family, and many other proteins involved in oncogenic transduction pathways [63]. The participation of HSP90 client proteins in the network of signal transduction pathways involved in tumor growth is highly complicated (Figure 3). Interestingly, HSP90 client proteins contribute to hallmarks of cancer, such as self-sufficiency in growth signaling, an insensitivity to antigrowth signals, the evasion of apoptosis, persistent angiogenesis, tissue invasion, metastasis, and an unbounded capacity for replication [64]. As a result, HSP90 appears to be a special molecular target, since its blocking would prevent all of the crucial pathophysiological processes that tumor cells rely on to grow and survive.

There is evidence from both experimental and clinical investigations suggesting that the epidermal growth factor (EGF) plays a significant role as an oncogenic growth factor in cancer [65]. Human carcinomas frequently overexpress EGF and its receptor, EGFR, and the human epidermal growth receptor-2 (HER-2/neu). The EGFR system and HER-2/neu both appear to be crucial in the control of tumor development and angiogenesis, since such expression has been linked to a negative prognosis for patients [66]. Importantly, the activation of oncogenic signaling pathways, such as phosphatidylinositol 3-kinase (PI3K)/Akt and mitogen-activated protein kinase (MAPK, also known as rat sarcoma (Ras)–rapidly accelerated fibrosarcoma (Raf)–MEK)/extracellular signal-regulated kinase (ERK), can in cancer cells lead to the induction of vascular endothelial growth factor-A (VEGF-A), one of the most potent proangiogenic factors [67]. The EGF/EGFR system is a regulator of these signaling intermediaries. Additionally, the transcription factor hypoxia-inducible factor-1 (HIF-1) may be activated by both hypoxia and cancer to produce VEGF-A, promoting the development and spread of tumors. HIF-1 has previously been shown to be an essential regulator of human cancer development and angiogenesis, making it important to target this protein [68]. However, the expression of HSP90 has also been linked to the onset of cancer and lymph node metastases; as a result, HSP90 has become a viable target for the treatment of cancer.

The HSP90 chaperoning function is crucial to the activity of the aforementioned carcinogenic proteins, including mutant EGFR, HER-2, MAPK/ERK, Akt, and transcription factor HIF-1. In non-small cell lung cancer, gastric cancer, and breast cancer, HER-2 and EGRF expression levels were decreased by the ganetespib-mediated suppression of HSP90 [69,70,71]. Ganetespib treatment may suppress the EGFR, which in turn may inhibit VEGF/VEGFR [72].

Ganetespib has shown antiangiogenic efficacy in colorectal cancer [72] and breast cancer [73] through HIF-1 suppression. The signaling of PI3k, MAPK, and the signal transducer and activator of transcription (STAT3) would be further suppressed by the inhibition of VEGFR. In non-small cell lung carcinoma and gastric cancer, a ganetespib treatment was previously found to have a suppressive impact on PI3K/Akt expression [73,74]. Ganetespib reduced signaling through the STAT3, PI3K/Akt, and MAPK pathways, which had the effect of slowing the proliferation of mutant pancreatic cells [75]. 

Chronic inflammation appears to be critical in the initiation and development of numerous human carcinomas, in addition to growth factor signaling. Chronic and persistent inflammation is very strongly related to malignant lesions. Interleukin-8 (IL-8) is one of the cytokines linked to malignancies. Ganetespib’s ability to disrupt HSP90 has recently been shown to reduce the generation of IL-8 [76]. This may be linked to STAT3/nuclear factor-κB (NF-κB) signaling modulation [77], which implies that the inhibition of HSP90 by ganetespib might reduce the pro-inflammatory cytokine signaling that is evoked in cancer.

Kristen rat sarcoma virus (KRAS oncogene) mutations are often seen in human cancer, and ganetespib has been shown to effectively inhibit HSP90, affecting the constitutive activation of several signaling transduction pathways, including Ras/Raf/MEK/ERK signaling components. In thyroid cancer, HSP90 suppression by ganetespib results in decreased Raf-1 expression [78] and the inhibition of both ERK1/2 and Akt phosphorylation, an effect that is also mirrored by a general decrease in Akt protein levels. Several survival mechanisms, including the PI3K pathway and signaling through NF-κB, depend on Akt signaling [79].

A number of human malignancies have prominent characteristics of tissue invasion and metastasis to distant organ locations. It was shown that HSP90 suppression by ganetespib impaired EGF- and HGF-induced signaling in human cancer cells, since the EGFR and c-mesenchymal–epithelial transition (c-Met) receptor (receptor to hepatocyte growth factor, HGF) systems both represent significant mediators of cancer development and metastasis. Additionally, ganetespib-induced HSP90 suppression significantly decreased the activation of the c-Met receptor by HGF [58,59,80] and the phosphorylation of focal adhesion kinase (FAK) [81].

However, in an in vivo model, the inhibition of HSP90 by ganetespib significantly decreased tumor growth and vascularization through a mechanism related to the tumor necrosis factor-related apoptosis-inducing ligand (TRAIL, also known as Apo2L), which was found to be a potent activator of apoptosis [82,83]. The cell functional death receptors (DR) TRAIL-R1 (DR4) and TRAIL-R2 (DR5) are ligated and trimerized to begin the apoptosis that TRAIL induces. Since radiation and well-established chemotherapeutic medicines have been demonstrated to work in synergy with TRAIL, it is a prospective option for cancer therapy, either alone or in combination.

Both the Akt and NF-B pathways may influence cytotoxic responses; however, the mechanisms behind either sensitivity or resistance have not yet been completely understood. It was proposed that inhibiting HSP90 would be effective for overcoming TRAIL resistance in colon cancer, whereas Akt or NF-B signaling may increase TRAIL resistance in cancer cells [84]. Another crucial point is that the inclusion of ganetespib may enhance the anti-neoplastic efficiency of oxaliplatin, a drug currently and often used in multimodality chemotherapy regimens for the treatment of advanced or metastatic (liver) colorectal cancer. Ganetespib altered PI3K/Akt/ERK signaling, sensitizing CRC cell lines to the effects of oxaliplatin [85]. Ganetespib may also suppress the G1-S transition in cancer cells by targeting the receptor tyrosine kinase, according to recent studies [74].

In the multimodal treatment plan for advanced human cancer, radiation therapy plays a significant role. The harm to the surrounding healthy tissues still restricts the administration of high doses of ionizing radiation (IR). Selectively sensitizing the tumor cells to IR and enabling the dosage reduction of IR in therapies is one potential answer to this issue. In this regard, recent studies have shown that the inhibition of a traditional MAPK pathway is a successful strategy for increasing the sensitivity of specific tumor cell types to IR therapy [86,87]. Most interestingly, it has been proven that the concurrent use of ganetespib has a radiosensitizing impact on cancers of the liver, colon, and breast, even at very moderate radiation doses [88,89,90]. Together, these several advantageous properties of ganetespib’s HSP90-targeting therapy for cancer emphasize the effectiveness and applicability of this cutting-edge molecular strategy and may serve as the foundation for further clinical research in this area.

Since ganetespib has the ability to block a variety of oncogenic signaling pathways, additional signaling pathways, including the STAT3 and insulin-like growth factor-I receptor (IGF-IR) signaling pathways, are thought to be involved. It was hypothesized that ganetespib’s HSP90-blocking effects on prostate cancer’s IGF-1- and IL-6-mediated signaling would prevent tumor development and angiogenesis. In fact, the ganetespib-mediated reduction of HSP90 prevented IGF-1R signaling by directly disrupting and downregulating IGF-IR [91,92]. Additionally, a ganetespib treatment significantly decreased the activation of HIF-1 or STAT3 caused by hypoxia and IL-6 [93].

**Figure 3 ijms-24-05014-f003:**
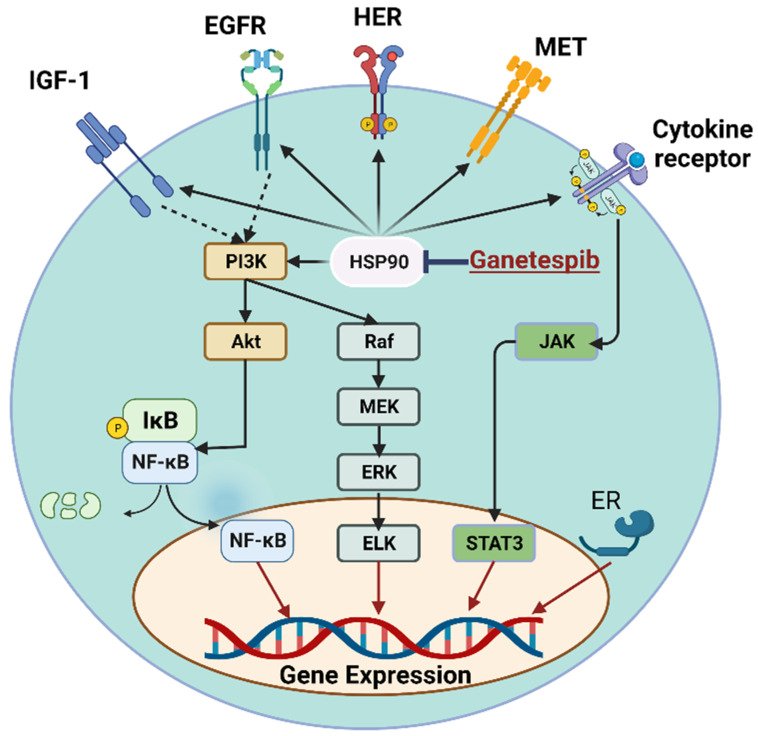
Molecular mechanisms of ganetespib. HSP90 increases the expression of IGF-1 [91,92], EGFR [72], HER [69,70,71], MET [58,59,80], ER [69], and cytokine receptors. Activation of IGF-1R and EGF stimulates PI3K/Akt/NF-κB and Raf/MEK/ERK signaling, respectively [85]. JAK/STAT3 signaling is enhanced by cytokine receptor activation [93]. Ubiquitinated degradation of the inhibitor of NF-κB (IκB), nuclear translocation of NF-κB, ERK, and STAT3 stimulate gene expression of mediators that increase tumorigenesis. Ganetespib-mediated inhibition of HSP90 leads to the suppression of these signaling mechanisms and anticancer effects.

## 10. Role of Ganetespib in Cancer Diseases

Ganetespib had strong activity in preclinical tests against a variety of cancer models (Table 3), including lung cancer, prostate cancer, and leukemia. These results indicated that a developing clinical profile would disclose signs of therapeutic effectiveness in human tumors, especially in non-small cell lung cancer [94], where ganetespib has demonstrated promising single-agent efficacy in molecularly defined subgroups of that disease. There has been a thorough assessment of ganetespib activity in breast cancer cell lines involving luminal and basal tissues, hormone receptor subtypes, HER2-positive cancer, TNBC (triple-negative breast cancer), and inflammatory breast cancer (IBC). According to the findings, ganetespib has a lot of potential as a different therapeutic approach and possibly as a complement to existing therapies for the treatment of breast cancer. Ganetespib is being tested in a global phase II study as a first-line therapy for metastatic breast cancer in light of these factors [69].

### 10.1. In Breast Cancer

Most human breast cancers are of the luminal type, which is characterized by high expression of the estrogen receptor (ER) and/or the progesterone receptor (PR). Ganetespib decreased the survival in two hormone receptor-positive cell lines with low nanomolar potency such as MCF-7 (Michigan cancer foundation-7) and T47D cells. Initially, and with 15 nM and 25 nM, T47D cells were used to examine how the ganetespib treatment affected receptor expression. When given ganetespib, both isoforms of PR (PR B and PR A) and ER exhibited a powerful and reliable dose-dependent destabilization. Next, in T47D cells, the kinetics of steroid receptor loss and pathway modification were investigated. After only six hours, a 250 nM ganetespib dose completely destabilized the PR and caused maximum ER level decreases. These effects persisted throughout 24 h. Ganetespib caused measurable reductions in PR and ER expression levels at 6 h at concentrations as low as 25 nM, and these effects increased stronger over time. Ganetespib 100 mg/kg was administered weekly to mice with MCF-7 xenografts, and these animals showed a substantial reduction in tumor volume. Therefore, ganetespib has strong anticancer activity in vivo by inhibiting the expression and survival of ER/PR proteins in hormone receptor-positive breast cancer cells [69].

The TNBC lines MDA-MB-231 and OCUB-M were sensitive to ganetespib, with IC50 values in the low nanomolar range. Therefore, ganetespib prevented TNBC cells from expressing oncogenic signals and growing tumors [69].

The uncommon, aggressive, and clinically different form of locally progressed breast cancer known as IBC is rare [95]. With an IC50 value of 13 nM, ganetespib proved highly cytotoxic to the well-studied IBC cell line SUM149. The biology underpinning IBC is still not fully understood, but multiple oncogenic signaling pathways are inhibited by ganetespib, which also exhibited targeted combinatorial activity in inflammatory breast cancer [69].

### 10.2. In Non-Small Cell Lung Cancer (NSCLC)

Studies have shown that ganetespib monotherapy demonstrates clinical activity in patients diagnosed with advanced NSCLCs who have received extensive prior therapy, particularly in those whose tumors have experienced an anaplastic lymphoma gene (ALK) rearrangement [96]. According to the results, a phase II investigation of ganetespib monotherapy in patients with crizotinib- naïve ALK-positive illness has just been started. In addition to further evaluating the use of several ALK testing methods (IHC, PCR, and FISH), this trial will characterize ganetespib activity in this population prospectively [96].

### 10.3. In Gastric Cancer (GC)

GC is still the fifth most common cancer in the world. Due to the abnormally high expression of HSP90 in malignancies, it has become a desirable therapeutic target in the fight against the disease. Ganetespib was tested because of several instances where HSP90 inhibitors were successful in suppressing GC. Studies have shown that ganetespib strongly suppressed the growth of MGC-803 and SGC-7901 as well as MKN-28 in a dose-dependent manner. In all three cell lines, it caused apoptosis and G2/M cell-cycle arrest [97].

### 10.4. In Acute Myeloid Leukemia (AML)

Ganetespib is considerably more potent than the conventional drug, cytarabine, against primary AML blasts at nanomolar dosages that are clinically feasible [98,99]. HSP90 inhibition has previously been found to have anti-proliferative effects in AML and other leukemias [98,100,101]. In addition, ganetespib has a significantly better efficacy in primary AML samples than earlier HSP90 medicines [98,101].

AML cells were found to have dose-dependent apoptosis, showing a cytotoxic method of cell death in response to treatment. Annexin-induced cell death occurred at somewhat higher drug dosages than those found in the MTS experiment, which could be attributed to ganetespib’s impact on the cell cycle regulator clients of HSP90. Ganetespib has already been established in various cancer models to cause growth arrest and death [92].

Because several protein kinases are duplicated in tumor maintenance, the effectiveness of any inhibitor may be dependent on oncogene addiction to the HSP90/client protein [92,102]. Ganetespib’s multi-client action allows it to inhibit many more sites than traditional kinase inhibitors, and when combined with existing chemotherapeutic and new medicines, it will be capable of targeting a wide range of molecular abnormalities, including those present in AML [98].

### 10.5. In Liver Cancer by Radiosensitization

The anti-tumor and radiosensitizing properties of ganetespib were investigated for the treatment of liver cancer. Ganetespib was able to decrease the activity of many client signal transduction proteins, including ERK1/2 and S6, in a dose-dependent manner, which is thought to be crucial for liver cancer cell survival and proliferation. It was discovered that ganetespib therapy alone decreased the clonogenic survival in all three of the hepatocellular carcinoma (HCC) cell lines examined and was also capable of radiosensitizing these HCC cells in vitro. Ganetespib therapy caused an increase in cytotoxicity in HCC cells. A possible mechanism for radiosensitization by ganetespib includes cell cycle disruptions that increase the proportion of cells in a G2-M arrest and decrease the levels of the double-strand break repair (DSBR) response protein CHK1. These observations were associated with a higher proportion of unrepaired DSBs after radiation exposure. At last, by utilizing a tumor xenograft model, it was demonstrated that ganetespib was capable of radiosensitizing HCC cells to fractionated radiation in vivo [90].

It was demonstrated that ganetespib can inhibit HSP90 in a dose-dependent manner, resulting in anti-cancer activity and the strong radiosensitization of HCC cell lines by affecting important cellular signaling pathways, causing cell cycle arrest in more radiosensitive phases of the cell cycle, and inhibiting DNA repair mechanisms. According to these results, ganetespib holds a promising role as a hepatic cancer treatment when properly paired with radiation, and clinical trials should be conducted to test this hypothesis [90].

### 10.6. In Metastatic Pancreatic Cancer

It has been demonstrated that several HSP90 client proteins are crucial for the development and maintenance of pancreatic cancer [1]. Ganetespib showed a satisfactory safety profile in phase I and phase II trials in patients with refractory metastatic or locally advanced solid tumors, as well as some effectiveness [99]. In light of these findings, the researchers created the current phase II trial with the supposition that ganetespib monotherapy would suppress HSP90 in patients with pancreatic cancer safely and effectively [103]. These studies have shown that with few side events, ganetespib monotherapy was well tolerated. There were no reported treatment-related deaths, and most toxicities were grade 1 or 2. The disease control rate after 8 weeks of treatment served as the study’s primary objective and was 28.6%. This trial was prematurely ended, since it did not achieve the pre-specified clinical efficacy according to the scheduled interim analysis. To be clear, the individuals in this trial who received ganetespib as a second- or third-line therapy had undergone extensive pretreatment [103].

### 10.7. In Refractory Metastatic Colorectal Cancer

Ganetespib was tested for safety and efficacy in 50 patients with metastatic colorectal cancer who were resistant to conventional treatments; however, there was no response. In patients with resistant metastatic colorectal cancer treated with this dose regimen, ganetespib did not result in tumor responses [104].

### 10.8. In Prostatic Cancer

In preclinical models of prostate cancer, studies have demonstrated that ganetespib has strong cytotoxic activity and antitumor effectiveness. Importantly, ganetespib therapy effectively promoted cancer cell death, regardless of the androgen sensitivity, because of its concurrent effects on oncogenic survival pathways and cell cycle progression. The evidence suggests that ganetespib may be a useful therapeutic option for prostate malignancies caused by androgen receptors, androgen-independent shortened versions of the receptor, and castration-resistant tumors that are no longer dependent on the receptor itself. A further analysis of this agent’s therapeutic value is required in light of these findings [92].

### 10.9. In Autosomal-Dominant Polycystic Kidney Disease

Autosomal-dominant polycystic kidney disease (ADPKD) is characterized by the progressive growth of renal cysts, kidney enlargement, hypertension, and, in most cases, end-stage renal disease [105]. In ADPKD, hereditary mutations impair the function of the polycystins (encoded by PKD1 and PKD2), leading to the loss of a cyst-repressive signal coming from the renal cilium [105]. Ciliary repair may be important for the pathogenesis of ADPKD, according to genetic research. The clients of heat-shock protein 90 (HSP90) include a variety of proteins involved in ciliary maintenance. Ganetespib reduced the proteasomal suppression of NEK8 and the Aurora-A activator trichoplein, inducing Aurora-A kinase activation and ciliary loss in vitro. The researchers used conditional mice models for ADPKD to conduct long-term (10 or 50 weeks) dosing tests showing that HSP90 inhibition produced long-term in vivo cilia loss, regulated cystic development, and alleviated symptoms caused by Pkd1 or Pkd2 loss. Therefore, in ADPKD, ganetespib inhibits ciliation and cystogenesis.

A glycolysis antagonist exhibited potential in the treatment of ADPKD, whereas 2-deoxy-d-glucose (2DG), a glycolysis inhibitor, demonstrated some success in regulating cystogenesis in quickly growing Ksp-Cre, Pkd1flox, and Pkd1 V/V mice models of polycystic kidney disease. However, combining ganetespib with 2-deoxy-d-glucose did not improve the efficacy because the administration of ganetespib, an independent HSP90 inhibitor, in combination with 2DG produced qualitatively similar effects in the reduction of cystogenesis and kidney growth [105].

**Table 3 ijms-24-05014-t003:** Effect of ganetespib on different types of cancer.

Cancer Type	Effect of Ganetespib	References
Breast Cancer	Decreases survival in hormone receptor-positive cell lines MCF-7 and T47D with low nanomolar potency. Destabilizes estrogen receptor (ER) and progesterone receptor (PR) in a dose-dependent manner. Shows strong anticancer activity in vivo by inhibiting expression and survival of ER/PR proteins in hormone receptor-positive cells. Prevents triple-negative breast cancer (TNBC) cells from expressing oncogenic signals and growing tumors. Proves highly cytotoxic to the inflammatory breast cancer (IBC) cell line SUM149.	[65]
Non-Small Cell Lung Cancer (NSCLC)	Demonstrated clinical activity in patients diagnosed with advanced NSCLCs who had received extensive prior therapy, particularly in those whose tumors had an anaplastic lymphoma gene (ALK) rearrangement. A phase II investigation of ganetespib monotherapy in patients with crizotinib-nave ALK-positive illness has been started.	[92]
Gastric Cancer (GC)	Suppresses the growth of MGC-803, SGC-7901, and MKN-28 in a dose-dependent manner. Causes apoptosis and G2/M cell-cycle arrest in all three cell lines.	[93]
Acute Myeloid Leukemia (AML)	More potent than the conventional drug, cytarabine, against primary AML blasts at nanomolar dosages. Causes dose-dependent apoptosis and annexin-induced cell death. Inhibits many more sites than traditional kinase inhibitors, making it capable of targeting a wide range of molecular abnormalities in AML.	[88,94,95,96,97]
Liver Cancer	Decreases the activity of client signal transduction proteins, including ERK1/2 and S6, in a dose-dependent manner. Decreases clonogenic survival in hepatocellular carcinoma (HCC) cell lines. Increases radiosensitivity in HCC cell lines.	[86]
Metastatic Pancreatic Cancer	Well tolerated with few side events and 28.6% disease control rate. Trial was prematurely ended due to lack of pre-specified clinical efficacy.	[1,95,99]
Refractory Metastatic Colorectal Cancer	No effect on tumor responses. No response in patients treated with ganetespib.	[100]
Prostatic Cancer	Strong cytotoxic activity and antitumor effectiveness. May be a useful therapeutic option for prostate malignancies caused by androgen receptors, androgen-independent versions of the receptor, and castration-resistant tumors. Further analysis required.	[88]
Autosomal-Dominant Polycystic Kidney Disease (ADPKD)	Inhibits ciliation and cystogenesis. Reduces proteasomal suppression and induces Aurora-A kinase activation and ciliary loss. Long-term dosing tests showed regulated cystic development and alleviated symptoms. Combination with 2-deoxy-d-glucose did not improve efficacy.	[101]

## 11. Ganetespib in Combination

HSP90 shields cells from stress; hence, HSP90 inhibitors can make cells more vulnerable to the harmful effects of chemotherapy and radiation therapy [55]. Preclinical evidence from several cell lines and xenograft models suggests additive or synergistic antitumor effects when HSP90 inhibitors are combined with various systemic cytotoxins, such as anthracyclines and taxanes [55,58,106].

The synergistic advantage of HSP90 inhibitors in combination with taxanes is likely the result of a number of factors, including increased cytotoxicity and apoptosis, Akt inactivation, the sensitization of the tumor cells to taxane-induced cell death, a decline in pro-survival signaling, and exacerbated mitotic catastrophe [58,106]. In fact, in TNBC, NSCLC, and ovarian cancer models, ganetespib and paclitaxel or docetaxel together promoted antitumor development and had synergistic effects [58,81]. Most critically, the toxicity profile of this mixture was non-overlapping. Ganetespib can also increase DNA damage and mitotic arrest, which can increase the cytotoxic activity of doxorubicin and provide doxorubicin-containing regimens with more effectiveness [58,81].

Tumor-associated KRAS mutations are found in 15–25% of lung adenocarcinomas. These mutations are poor predictors of responsiveness to currently available EGFR tyrosine kinase therapies, and patients with them have poor clinical outcomes [107]. There are no approved anti-KRAS-directed treatments at the moment. Although KRAS is not a known client protein by itself, HSP90 inhibition has been demonstrated to have an impact on KRAS’ downstream effector pathways, specifically the PI3K/AKT/mTOR and the RAF/MEK/ERK pathways. In fact, ganetespib plus a dual PI3K/mTOR inhibitor showed higher antitumor effectiveness in a xenograft model, prompting future research into this dual-targeted strategy [58,108].

Ganetespib was more effective than single-agent vemurafenib in melanoma cell lines driven by mutant v-raf murine sarcoma viral oncogene homolog B1 (BRAF), because this protein is a very sensitive HSP90 client. Since BRAF mutations cause the RAF/MEK/ERK signaling pathway to become dysregulated, several trials are looking at the effects of combining BRAF inhibitors and MEK inhibitors in order to increase efficacy and overcome acquired resistance. Similarly, xenograft models have demonstrated synergistic actions when ganetespib and vemurafenib, or specific MEK inhibitors, are combined; ganetespib and the MEK inhibitor TAK-733 together also significantly reduced the tumor size in vemurafenib-resistant xenograft tumors in a preclinical investigation, supporting the utility of this combination to treat cancers that have acquired a resistance to vemurafenib [109].

Ganetespib restores endocrine resistance and lowers heterogeneity in the disease control brought on by hormonal therapy in preclinical hormone receptor-positive breast cancer models [110]. This research has served as the foundation for assessing ganetespib interactions with endocrine treatments such as fulvestrant, a full-spectrum estrogen receptor antagonist.

The first medicinal proteasome inhibitor licensed by the Food and Drug Administration for the treatment of multiple myeloma and mantle cell lymphoma was bortezomib. According to preclinical research, the combination of ganetespib and bortezomib can increase the amount of bortezomib-triggered apoptosis and cause a prolonged intracellular accumulation of ubiquitinated proteins. This is attributed to the synergistic suppression of the 20S proteasome’s chymotryptic activity [58,111].

## 12. Side Effects of Ganetespib

Most of the toxicities caused by ganetespib were of severity grades 1 or 2. The most frequently reported adverse effect was diarrhea [96,112], which was easily treatable with loperamide or diphenoxylate atropine. Both geldanamycin and non-geldanamycin HSP90 inhibitors are associated with diarrhea [55,96]. Although it is unknown if this is an on-target impact, it might be related to the gut’s EGFR degradation [113]. Ganetespib-related investigations in the future will include prophylactic therapy for diarrhea [96].

With ganetespib, serious liver function test abnormalities were less common than with geldanamycins. Interestingly, ongoing preclinical analyses suggest that the physiochemical features of ganetespib likely lead to lower retinal/plasma concentrations and more effective retinal clearance than other HSP90 inhibitors, explaining the low incidence of ocular damage [96].

## 13. Conclusions

HSP90 synthesis is enhanced in cancer, inhibiting the disintegration of numerous client proteins. As a result, it aids in carcinogenesis and cancer metastasis by increasing cell survival and proliferation while lowering tumor cell death via regulating several client proteins, including HER-2, IGF-1R, ER, EGFR, and cytokine receptors. These pathways contribute to growth signaling, an insensitivity to antigrowth signals, the evasion of apoptosis, persistent angiogenesis, tissue invasion, metastasis, and an unconstrained capacity for the reproduction of these proteins. On the other hand, it is believed that one prospective strategy for the treatment of cancer involves utilizing ganetespib to inhibit HSP90 activity. Ganetespib inhibited growth signaling mechanisms and reduced cancer metastasis in several experimental and clinical studies. Further studies are needed to estimate tumor resistance against ganetespib.

## Figures and Tables

**Figure 1 ijms-24-05014-f001:**
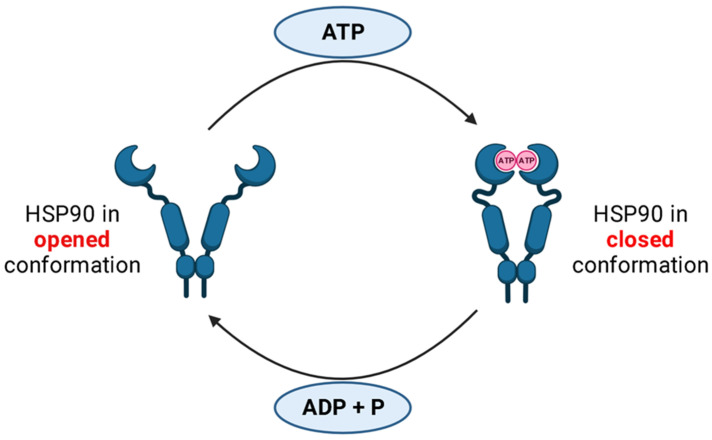
ATPase activity of HSP90. The HSP90 ATPase cycle is shown schematically. HSP90 is shown in both its open and closed conformations. The three structural domains NTD, MD, and CTD constitute the monomers of the flexible homodimer HSP90. A series of conformational events are triggered when ATP binds to its binding cleft in HSP90’s NTD. The formation of a twisted and compressed dimer is caused by the translocation of a short N-domain fragment (ATP-lid) over the binding pocket and the subsequent attachment to the corresponding N-domain of the opposing homodimer. The N- and M-domains then shift closer to one another, completing the “split ATPase” site. Following ATP hydrolysis, the N-domains of the HSP90 homodimers split apart, releasing ADP and Pi while the HSP90 returns to its initial open conformation [29,30,31].

**Figure 2 ijms-24-05014-f002:**
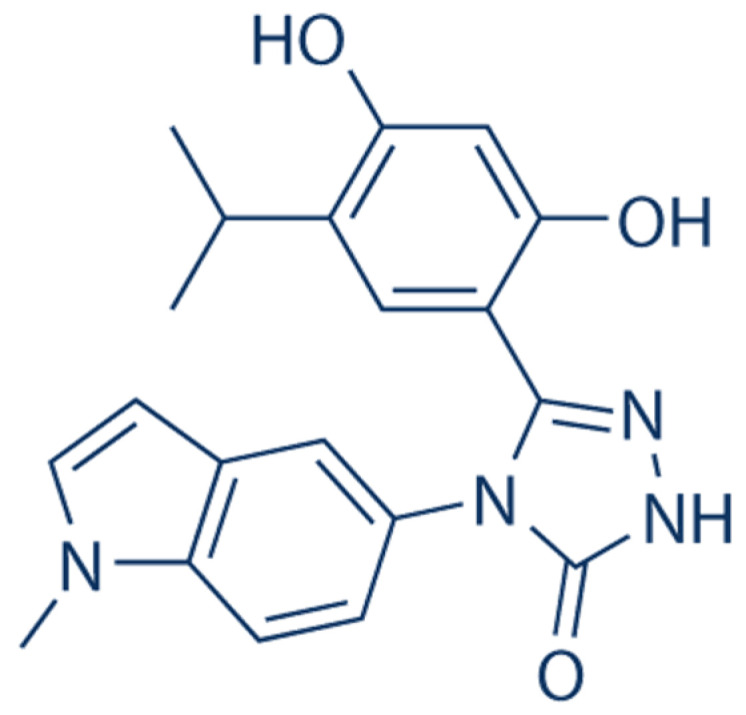
Chemical structure of ganetespib. Ganetespib, 5-[2,4-dihydroxy-5-(1 methylethyl)phenyl]-2,4 dihydro-4-(1-methyl-1H indol-5 yl)-3H-1,2,4 triazole-3-one, is a powerful, small-molecule inhibitor of HSP90 that attaches to the ATP pocket in the N-terminus of HSP90, thus leading to a decrease in the levels of HSP90 customer proteins. It has a distinct triazolone-containing chemical structure and is distinct from other HSP90 inhibitors in terms of its potency, anti-tumor action, and safety [59].

**Table 1 ijms-24-05014-t001:** Classification of heat-shock proteins (HSPs).

Family	Members	Characteristics	Functions	References
HSPA (HSP70)	13	The most commonly studied include HSPA1A and HSPA1B, which are functionally interchangeable and heat-inducible. HSPA6, HSPA8 (Hsc70 or HSP73), and HSPA6 are heat-induced. HSPA8 is essential and is constitutively expressed. HSPA7 is transcribed and seems to be induced by heat at least, but there is no evidence there is a protein product associated with it. HSPA5 is found in the ER, is constitutively expressed, and is upregulated by the unfolded protein response (UPR). HSPA1L and HSPA2 are highly expressed in the testes and are constitutively expressed. HSPA9 (mortalin/mtHSP70/GRP75/PBP74) is constitutively expressed and upregulated by the unfolded protein response (UPR). HSPA12A, HSPA12B, and HSPA14 are more distantly related members for which very little data is available.	Co-translational folding; protein translocation across membranes	[10,11]
HSPH (HSP110)	4	The HSPH family (four members) has a longer linker domain between the N-terminal ATPase domain and the C-terminal peptide-binding domain.	Nucleotide exchange factor for HSPA family	[10,12]
DNAJ (HSP40)	4 Type A 14 Type B 22 Type C	The DNAJ (HSP40) family is one of the largest HSP families present in humans. Type A proteins contain an N-terminal J-domain, a glycine/phenylalanine-rich region, a cysteine-rich region, and a variable C-terminal domain. Type B proteins possess an N-terminal J-domain and an adjacent glycine/phenylalanine-rich region, and the most widely expressed and heat-inducible human DNAJ member is DNAJB1. Type C proteins only have a J-domain, which may help recruit HSPA members to specific sub-compartments and/or functions.	Recruitment of HSPA members; protein folding and assembly	[10]
HSPB (small HSP)	11	The best-studied members are HSPB1 (HSP27), HSPB4 (αA crystallin), and HSPB5 (αB crystallin). These small HSPs are often found in oligomeric complexes involving one or more family members, providing the cell with diverse chaperone specificity. High expression of many members is found in skeletal and cardiac muscles, as well as many other tissues.	Protein folding and assembly; oligomeric complexes	[10,13]
HSPC (HSP90)	5	The HSPC (HSP90) family encodes five members, which were initially annotated as HSPC members, including: HSPC1 (HSP90AA1; HSPN; LAP2; HSP86; HSPCA; HSP89; HSP90; HSP90A; HSP90N; HSPCAL1; HSPCAL4; FLJ31884). HSPC2 (HSP90AA2; HSPCA; HSPCAL3; HSP90ALPHA). HSPC3 (HSP90AB1; HSPC2; HSPCB; D6S182; HSP90B; FLJ26984; HSP90-BETA). HSPC4 (HSP90B1; ECGP; GP96; TRA1; GRP94; endoplasmin). HSPC5 (TRAP1; HSP75; HSP90L).	Protein folding and assembly	[10,12]
HSPD/E	1	Single orthologs of *E. coli* GroEL (HSP60) and GroES (HSP10).	Protein folding and assembly	[14]
CCT (TRiC)	8	In the cytosol, CCT (TRiC) is a heterooligomeric chaperonin complex made up of 8 subunits. These subunits are encoded by separate genes with approximately 30% amino acid sequence identity (15–20% to GroEL). Two genes, CCT6A and CCT6B, encode CCT6 (zeta subunit). Eight genes of this family are annotated as CCT2–CCT5, CCT6A, CCT6B, CCT7, and CCT8.	Folding newly synthesized proteins; preventing protein aggregation	[10,15]
Chaperonin-like	3	Three chaperonin-like genes, MKKS/BBS6, BBS10, and BBS12, have been identified and are linked to McKusick–Kaufman syndrome and Bardet–Biedl syndrome.	Cilia and centrosome/basal body functions	[16]
HSP47	1	ER-resident protein that is part of the serine peptidase inhibitor (serpin) family.	Collagen-specific chaperone	[17]

**Table 2 ijms-24-05014-t002:** Hsp90 inhibitors.

Class	HSP90 Inhibitor	Description	References
Natural products	Geldanamycin	Binds to the N-terminal ATP-binding site of HSP90, leading to client protein degradation and interference with cellular processes. Not suitable for clinical use due to toxicity, instability, and poor solubility.	[1]
	Radicicol	Binds to the ATP-binding site of HSP90, demonstrating potential as a multi-faceted approach to treating cancer.	[42]
Semisynthetic	17-AAG	Geldanamycin derivative that entered phase I clinical trials, but did not advance past phase II. Poor bioavailability.	[43,44]
	17-AAGH2	Improved the metabolic profile and potency of 17-AAG. Clinical trials were terminated or withdrawn.	[21]
	17-DMAG	Developed to improve the physiochemical profile of geldanamycin.	[45]
Purine-based	PU-3	First fully synthetic small-molecule HSP90 inhibitor; inhibits growth and differentiation in breast cancer cells.	[46]
	BIIB021 BIIB028 MPC-3100 PU-H71 Debio0932	Derivatives of the purine chemical scaffold in PU-3; entered clinical trials, but did not advance past phase II.	[47]
Benzamide	SNX-5422	Pyrazole-containing HSP90 inhibitor with promising bioavailability, but inhibits all HSP90 isoforms, leading to dose escalation challenges.	[48]
Resorcinol-containing	AUY922	Resorcinol-containing HSP90 inhibitor; active against tumor growth, angiogenesis, and metastasis.	[49]
	STA-9090	Modification of the resorcinol scaffold with a higher potency in downregulating oncoproteins and pathways. Advanced to phase III trials and is the most clinically evaluated HSP90 inhibitor.	[50]
	AT13387	Resorcinol-containing HSP90 inhibitor discovered by Astex Therapeutics, with a high potency and ideal ligand efficacy.	[51]
	KW-2478	Discovered by scientists at Kyowa Hakko Kirin in Japan.	[51]
	Ganetespib	Second-generation HSP90 inhibitor composed of the resorcinol moiety of radicicol derivatives.	[51]
Miscellaneous	XL888 HSP990 DS2248 TAS-116	Miscellaneous inhibitors, except for TAS-116, all work by binding to the NTD of HSP90.	[52,53]

## Data Availability

Not applicable.

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
