# Peer review of "Role of Ganetespib, an HSP90 Inhibitor, in Cancer Therapy: From Molecular Mechanisms to Clinical Practice"

_ijms, 2023, doi:10.3390/ijms24055014_

Round 1

Reviewer 1 Report

Exploring the role of Ganetespib, an HSP90 inhibitor, in Cancer Therapy seems to be very interesting, yet some concerns should be addressed to improve the quality of the review:

1- The fluency of the English language can be enhanced with careful consideration for grammar and punctuation.

2- The authors should discuss the correlation between HSP90 and HSP70 as there is a strong correlation between them

3- Does Ganetespib have any impact on the levels of HSP70?

4- The safety of Ganetespib also should be discussed as the author suggest its clinical use.

5- More details about the pharmacokinetic profile, distribution, elimination, and other relevant aspects of the drug will be beneficial.

6- Examine the effect of Ganetespib on cell death pathways, such as apoptosis, macroautophagy, chaperone-mediated autophagy, necroptosis, etc. and other cancer related pathways.

7- The authors should expand the introduction by addressing previous concerns surrounding the topic.

8- The conclusion section should be revised to provide a more comprehensive summary of the findings.

Author Response

Thank you for considering this manuscript for publication in the International Journal of Molecular Sciences. The authors would like to thank the reviewers for their valuable comments that helped us to present the current study in a better way. All points addressed by the reviewers were answered and every addition/modification is highlighted in the manuscript whenever possible. I hope that our replies will be satisfactory. We look forward to your positive feedback.

Comment

1- The fluency of the English language can be enhanced with careful consideration for grammar and punctuation.

Response

Enhancements are done and the manuscript is amended in terms of grammar and punctuation

Comment

2- The authors should discuss the correlation between HSP90 and HSP70 as there is a strong correlation between them

Response

We briefly pointed to the relationship in section ‘’Types of heat shock proteins (HSP)’’ 

Comment

3- Does Ganetespib have any impact on the levels of HSP70?

Response

Yes, a general notification presents in section ‘’Types of heat shock proteins (HSP)’’ 

Comment

4- The safety of Ganetespib also should be discussed as the author suggest its clinical use

Response

Please revise section 10.6

Comment

5- More details about the pharmacokinetic profile, distribution, elimination, and other relevant aspects of the drug will be beneficial.

Response

We believe that these data are not relevant to the scope and will be redundant. Thank you for this advise

Comment

6- Examine the effect of Ganetespib on cell death pathways, such as apoptosis, macroautophagy, chaperone-mediated autophagy, necroptosis, etc. and other cancer related pathways.

Response

Precious information are scattered allover the manuscript and no need to collect under separate section

Comment

7- The authors should expand the introduction by addressing previous concerns surrounding the topic.

Response

The introduction is improved

Comment

8- The conclusion section should be revised to provide a more comprehensive summary of the findings.

Response

The conclusion is amended according to your recommendation and other reviewers

Reviewer 2 Report

The article of Mahmoud E. Youssef et al. entitled " Role of Ganetespib, an HSP90 inhibitor, in Cancer Therapy: From Molecular Mechanisms to Clinical Practice" describes the role of heat shock protein HSP90 in the development of the neoplastic process and how its inhibition by a small molecule compound - ganetespib, may have a beneficial effect in anticancer therapies. It is a well-structured and very interesting article. This description is very detailed and supported by the latest scientific literature. The 46 references (out of 70) come from the last decade, most of them from the last few years. In my opinion, the authors have made a reliable review of the literature on this topic. Unfortunately, the work contains several errors that should be corrected. They are listed below.

1. In Figure 2, the incorrect formula ganetespib is given. Ganetespib contains a triazolone and indole ring in its structure.

2. The text in lines 166-173 and 220-226 is duplicated in a given paragraph.

3. The list of Abbreviations is missing several abbreviations used in the text, such as e.g. ALK, ELK, IκB, MET, MEK, Raf.

4. The terms in vitro and in vivo should be in italics.

5. The name ganetespib is sometimes capitalized (e.g. in lines 108, 131, 236).

6. In the line 342 there is „grade 1 or 2” and in the line 427 there is „grades II or I”. I think that should be unified.

7. There are some editorial errors:

·         dot before reference to literaturÄ™ (e.g. in lines 86, 91, 100, 270, 398),

·         no spaces, especially before references to literature (e.g. in lines 57, 105, 252, 275, 295, 383, 404, 425, 431),

·         in the line 396 there is „15%-25%” and it should be „15-25%”,

·         in the line 373 before Ganetespib there is a dot and I think it should be a comma,

·         in the lines 377 and 378 words „Combining” and „The” should be written in lowercase.

8. There are some typos errors:

·         in the line 99 there is „Ganetesbip” and it should be „Ganetespib”,

·         in the line 350 there is „no respondents” and it should be „no respond”,

·         in the line 452 there is „amd” and it should be „and”.

Author Response

Thank you for considering this manuscript for publication in the International Journal of Molecular Sciences. The authors would like to thank the reviewers for their valuable comments that helped us to present the current study in a better way. All points addressed by the reviewers were answered and every addition/modification is highlighted in the manuscript whenever possible. I hope that our replies will be satisfactory. We look forward to your positive feedback.

Comment

1. In Figure 2, the incorrect formula ganetespib is given. Ganetespib contains a triazolone and indole ring in its structure.

Response

Thanks for your comment, the correct formula is added in figure 2.

Comment

2. The text in lines 166-173 and 220-226 is duplicated in a given paragraph.

Response

The duplication was deleted in line 243.

Comment

3. The list of Abbreviations is missing several abbreviations used in the text, such as e.g. ALK, ELK, IκB, MET, MEK, Raf

Response

All abbreviation were added and highlighted in the main text and in the abbreviation list.

Comment

4. The terms in vitro and in vivo should be in italics.

Response

All terms were modified into italic formats. 

Comment

5. The name ganetespib is sometimes capitalized (e.g. in lines 108, 131, 236)

Response

We corrected the capitalized words in lines 108, 131, 236.

Comment

6. In the line 342 there is „grade 1 or 2” and in the line 427 there is „grades II or I”. I think that should be unified.

Response

We unified the numbering format into Arabic numbers.

Comment

7. There are some editorial errors:

·         dot before reference to literaturÄ™ (e.g. in lines 86, 91, 100, 270, 398),

·         no spaces, especially before references to literature (e.g. in lines 57, 105, 252, 275, 295, 383, 404, 425, 431),

·         in the line 396 there is „15%-25%” and it should be „15-25%”,

·         in the line 373 before Ganetespib there is a dot and I think it should be a comma,

·         in the lines 377 and 378 words „Combining” and „The” should be written in lowercase.

Response

All these editorial errors were corrected

Comment

8. There are some typos errors:

·         in the line 99 there is „Ganetesbip” and it should be „Ganetespib”,

·         in the line 350 there is „no respondents” and it should be „no respond”,

·         in the line 452 there is „amd” and it should be „and”.

Response

All these errors were corrected

Reviewer 3 Report

In manuscript IJMS-2204039, Youssef M. and colleagues summarized current studies on hsp90 inhibitor Ganetespib, from mechanism to medical practice in different cancers. The layout of the manuscript is clear and in order but not in depth. The references from this study were out-dated and missed many current reports on Ganetespib. Overall, this manuscript needs improvement to meet the standard to publish in IJMS. Here are the specific comments:

(1) The authors need to rewrite the abstract to focus on the ganetespib, to summarize its potential mechanism and medical practice. The current abstract wrote too much about Hsp90 client proteins and signaling pathways.

(2) In Table 1, authors did not list all the HSP proteins. The source of this table was from one single outdated report.

(3) In Figure 1, authors should put client proteins into the illustration for better understanding the cycle.

(4) Since the manuscript stated Ganetespib was second generation Hsp90 inhibitor, authors should briefly introduced the first generation hsp90 inhibitor and their differences.

(5) Line 43, cellular and internal was redundant.

(6) Line 137-142, the description about EGFR and HER2 were not correct. Authors should read more on EGFR family and rewrite the statement.

(7) Line 17-18, the abbreviation of EGFR, HER and IGF1 were wrong.

(8) Line 166-173, texts were duplicated.

(9) Figure 3, IGF1R would be the right name instead of IGF1, please check the whole text about IGF1.

(10) Line 442, the client proteins should be HER2, IGF1R.

(11) Line 443, include should be removed.

Author Response

Thank you for considering this manuscript for publication in the International Journal of Molecular Sciences. The authors would like to thank the reviewers for their valuable comments that helped us to present the current study in a better way. All points addressed by the reviewers were answered and every addition/modification is highlighted in the manuscript whenever possible. I hope that our replies will be satisfactory. We look forward to your positive feedback.

Comment

(1) The authors need to rewrite the abstract to focus on the ganetespib, to summarize its potential mechanism and medical practice. The current abstract wrote too much about Hsp90 client proteins and signaling pathways.

Response

We have rewritten the abstracted and we added  several sentences that describe the medical practice of ganetespib.

Comment

(2) In Table 1, authors did not list all the HSP proteins. The source of this table was from one single outdated report.

Response

Thank you for pointing out the missing information in Table 1. We apologize for the oversight. We took your feedback into consideration and made sure to update the table to include all relevant HSP proteins. Additionally, we reviewed the source of the information and made sure it is up-to-date and reliable.

Comment

(3) In Figure 1, authors should put client proteins into the illustration for better understanding the cycle.

Response

We illustrated of ATPase activity on HSP90 in more details in figure 1 caption to understand the cycle. However, the effect of HSP90 on other clients proteins is described in details in figure 3.

Comment

(4) Since the manuscript stated Ganetespib was second generation Hsp90 inhibitor, authors should briefly introduced the first generation hsp90 inhibitor and their differences.

Response

We added a section (section 5) that describes the history of the discovery of HSP90 inhibitors

Comment

(5) Line 43, cellular and internal was redundant

Response

We corrected this mistake

Comment

(6) Line 137-142, the description about EGFR and HER2 were not correct. Authors should read more on EGFR family and rewrite the statement

Response

Thank you for bringing the inaccuracies in our description of EGFR and HER2 to our attention. We appreciate your grateful for the opportunity to improve the accuracy of our work. We took your suggestion and read more on the EGFR family to ensure that our statement is correct and up-to-date.

Comment

(7) Line 17-18, the abbreviation of EGFR, HER and IGF1 were wrong

Response

We corrected the abbreviation of these terms.

Comment

(8) Line 166-173, texts were duplicated.

Response

We removed the duplication .

Comment

(9) Figure 3, IGF1R would be the right name instead of IGF1, please check the whole text about IGF1.

Response

The abbreviation of insulin like growth factor -1 receptor is abbreviated as IGF-1R in the whole manuscript.

Comment

(10) Line 442, the client proteins should be HER2, IGF1R

Response

We have corrected this mistake.

Comment

(11) Line 443, include should be removed

Response

This mistake was corrected.

Author Response

Thank you for considering this manuscript for publication in the International Journal of Molecular Sciences. The authors would like to thank the reviewers for their valuable comments that helped us to present the current study in a better way. All points addressed by the reviewers were answered and every addition/modification is highlighted in the manuscript whenever possible. I hope that our replies will be satisfactory. We look forward to your positive feedback.

Comment

what certainly needs to be improved is the English. There are grammatical errors, typos, incomplete and meaningless sentences throughout the manuscript. Even in the first line of the abstract ‘Heat shock protein 90’ is written as ‘heat shock protein’ and ‘clients’ is written as ‘clints’. There are similar errors throughout the manuscript, and they are too many to list. This spoils the enthusiasm of the readers, unfortunately. I strongly suggest that the manuscript should be thoroughly read and corrected (or re-written in some places) preferably by a native English speaker. Otherwise, I am okay with the content.

Response

Thank you for your feedback. We appreciate your concern about the English in the manuscript and agree that it needs improvement. We will take your suggestions into consideration and make sure to have the manuscript thoroughly read and corrected, with a focus on fixing grammatical errors, typos, and ensuring complete and meaningful sentences. Your suggestions will help us improve the quality of the manuscript and enhance the reader's experience. Thank you for taking the time to provide this constructive criticism.

Reviewer 5 Report

The Review describes the role of HSP90 inhibitor Ganetespib in cancer. It is interesting Review, however, there are several concerns that should be addressed: 

1)     In the Introduction, authors should include several sentences on all HSP families

2)     Table 1 is incorrect. 1)  Authors used incorrect classification for the nomenclature of HSPs. Authors should use the nomenclature by Kampinga, et al, 2009 https://doi.org/10.1007/s12192-008-0068-7 2) HSP70 or HSP90 are families. Each family consists of several homologs and not all these homologs present in the same compartment.. Authors should refer to the nomenclature and do extensive literature review and then re-do the Table 1 accordingly

3)     Authors did not include information on HSP90 family…Please include separate section on HSP90 family, structure and functions

4)     There is also no information in the manuscript on the role of extracellular HSP90 in cancer.. Please include separate section on extracellular HSP90 in cancer

5)     Please include Table with all clinical trials that assessed the efficacy of Ganetespib in different types of cancer

6)      Please include the section on Other HSP90 inhibitors and their mechanism of action

7)     In the Abstract, authors described signalling pathways that are mainly associated with HSP90 rather than with HSPs

8)     Line 59 , GroEL/ES – these are bacterial HSPs. May be it will be more appropriately to write HSP60 and HSP10?

Minor comments

Line 15 – please change “clints” to “clients”

Author Response

Thank you for considering this manuscript for publication in the International Journal of Molecular Sciences. The authors would like to thank the reviewers for their valuable comments that helped us to present the current study in a better way. All points addressed by the reviewers were answered and every addition/modification is highlighted in the manuscript whenever possible. I hope that our replies will be satisfactory. We look forward to your positive feedback.

Comment

1)     In the Introduction, authors should include several sentences on all HSP families

Response

We added a section (section 2) that illustrates the classification of HSPs

Comment

2)     Table 1 is incorrect.

1)  Authors used incorrect classification for the nomenclature of HSPs. Authors should use the nomenclature by Kampinga, et al, 2009 https://doi.org/10.1007/s12192-008-0068-7 

2) HSP70 or HSP90 are families. Each family consists of several homologs and not all these homologs present in the same compartment.. Authors should refer to the nomenclature and do extensive literature review and then re-do the Table 1 accordingly

Response

We have summarized the HSP families in table 1.

Comment

3)     Authors did not include information on HSP90 family…Please include separate section on HSP90 family, structure and functions

Response

We added section 3 in the main manuscript. It gives a brief description of HSP90 family members and their main functions

Comment

4)     There is also no information in the manuscript on the role of extracellular HSP90 in cancer. Please include separate section on extracellular HSP90 in cancer

Response

We added a section (section 6) that illustrates the role of extracellular HSP90 (eHSP90) in cancer.

Comment

5)     Please include Table with all clinical trials that assessed the efficacy of Ganetespib in different types of cancer

Response

We included a table (Table 2) the describes clinical and experimental effects of ganetespib

Comment

6)      Please include the section on Other HSP90 inhibitors and their mechanism of action

Response

The classification of HSP90 inhibitors and their mechanism were added in section 7.

Comment

7) In the Abstract, authors described signalling pathways that are mainly associated with HSP90 rather than with HSPs

Response

The review focuses mainly on the effect of ganetespib, a selective HSP90 inhibitor, on the treatment of cancer. So we focused on HSP90 as a target of ganetespib and its role in cancer. Other HSPs were described briefly in section 2.

Comment

8)     Line 59 , GroEL/ES – these are bacterial HSPs. May be it will be more appropriately to write HSP60 and HSP10?

Response

We changed it into HSP60 and HSP10

Comment

Line 15 – please change “clints” to “clients

Response

We have corrected this mistake

Round 2

Reviewer 2 Report

I thank the Authors for making the corrections I suggested. However, the ganetespib formula in Figure 2 is still incorrect. I think it is a serious error if the formula of the compound under study is incorrect. The correct chemical formula of ganetespib can be found for example at https://www.selleckchem.com/products/ganetespib-sta-9090.html (CAS No. 888216-25-9).

Author Response

Thank you for considering this manuscript for publication in the international journal of molecular science. The authors would like to thank the reviewers for their valuable comments that helped us to present the current study in a better way. All points addressed by the reviewers were answered and every addition/modification is highlighted in the manuscript whenever possible. I hope that our replies will be satisfactory. We look forward to your positive feedback.

Comment

I thank the Authors for making the corrections I suggested. However, the ganetespib formula in Figure 2 is still incorrect. I think it is a serious error if the formula of the compound under study is incorrect. The correct chemical formula of ganetespib can be found for example at https://www.selleckchem.com/products/ganetespib-sta-9090.html (CAS No. 888216-25-9).

Response

authors have already made corrections based on your suggestions. The link you provided was a helpful resource for verifying the correct chemical formula of ganetespib. Thank you for bringing attention to this issue and promoting accuracy in this research.

Reviewer 3 Report

All the points were addressed by authors.I recommend for acceptance of this manuscript. 

Author Response

Thank you for considering this manuscript for publication in the international journal of molecular science. The authors would like to thank the reviewers for their valuable comments that helped us to present the current study in a better way. All points addressed by the reviewers were answered and every addition/modification is highlighted in the manuscript whenever possible. I hope that our replies will be satisfactory. We look forward to your positive feedback.

Comment

All the points were addressed by authors.I recommend for acceptance of this manuscript.

Response

It's great to hear that all the points raised were addressed by the authors. Thank you for your recommendation and for your contribution to the peer-review process.

Reviewer 5 Report

Authors have improved the manuscript, however, there are still major concerns:

1) lines 55-66 and Table 1 -information here is not correct. Please use classification by Kampinga et al “guidelines for the nomenclature of heat shock proteins…”. Please re-write Lines 55-66 and Table 1. The information present there is not correct.

2) lines 90-94 also not correct. There is no TRAP2 in HSP90 family. 

3) Authors should check the manuscript on the scientific information that authors provide here. Please re-check the manuscript according to the scientific literature. Please use Kampinga, et al, 2009 for the classification of HSP90 members present in the cell. 

4) All information provided under figures should be cited. It is not clear where to look for the information present under the Figure 1 and Figure 3. Please cite all information that you provided under all Figures.

5) figure 2 - no explanation under the Figure. Please provide the explanation under the Figure 2, citing the appropriate source. 

 6) Section 6: Roles of extracellular HSP90 - there are no information on the extracellular forms of HSP90( membrane, extracellular vesicles, etc). Authors also did not provide any mechanisms by which HSPs are exported outside the cells..

Author Response

Thank you for considering this manuscript for publication in the international journal of molecular science. The authors would like to thank the reviewers for their valuable comments that helped us to present the current study in a better way. All points addressed by the reviewers were answered and every addition/modification is highlighted in the manuscript whenever possible. I hope that our replies will be satisfactory. We look forward to your positive feedback.

Comment

1) lines 55-66 and Table 1 -information here is not correct. Please use classification by Kampinga et al “guidelines for the nomenclature of heat shock proteins…”. Please re-write Lines 55-66 and Table 1. The information present there is not correct.

Response

Thanks for you comment, we have rewritten the whole section and table 1 was completely altered based on information retrieved from Kampinga et al “guidelines for the nomenclature of heat shock proteins…”.

Comment

2) lines 90-94 also not correct. There is no TRAP2 in HSP90 family.

Response

We apologize for this mistake, we have deleted this information from the text.

Comment

3) Authors should check the manuscript on the scientific information that authors provide here. Please re-check the manuscript according to the scientific literature. Please use Kampinga, et al, 2009 for the classification of HSP90 members present in the cell.

Response

We checked the manuscript carefully and have read the article entitled “guidelines for the nomenclature of heat shock proteins” Kampinga et al.. The classification of HSP90 members was added in table 1

Comment

4) All information provided under figures should be cited. It is not clear where to look for the information present under Figure 1 and Figure 3. Please cite all information that you provided under all Figures.

Response

We have added citations in all figure captions

Comment

5) figure 2 - no explanation under the Figure. Please provide the explanation under the Figure 2, citing the appropriate source

Response

The description of figure 2 was added and cited

Comment

 6) Section 6: Roles of extracellular HSP90 - there are no information on the extracellular forms of HSP90 (membrane, extracellular vesicles, etc). Authors also did not provide any mechanisms by which HSPs are exported outside the cells..

Response

Thank you for your comment. We appreciate your feedback and agree that further information on the mechanisms by which HSP90 is exported outside the cells and the different forms of extracellular HSP90 would enhance the completeness of our discussion. To address these points, we have added a paragraph in the Roles of extracellular HSP90 section that provides a more detailed description of the mechanisms of HSP90 secretion and the different forms of extracellular HSP90, including membrane bound HSP90, exosomal HSP90, and secreted HSP90. We hope this additional information will help to clarify the role of extracellular HSP90 in cellular physiology and pathology. Thank you again for your input.

Round 3

Reviewer 2 Report

I believe that the manuscript in this form is fit for publication in IJMS.

Author Response

We greatly thank the reviewers for their time and effort during the revision and for their fruitful comments to improve our manuscript, herein we provide point to point report to the reviewer comments, and we highlighted all the changes in the revised version of the manuscript.

I believe that the manuscript in this form is fit for publication in IJMS.

  • We thank the reviewer for his nice comment, and we hope that our manuscript gives more impact to IJMS and the readers interested in HSP proteins as therapeutic targets for cancer

Reviewer 5 Report

The authors have addressed all the comments 

Author Response

We greatly thank the reviewers for their time and effort during the revision and for their fruitful comments to improve our manuscript, herein we provide point to point report to the reviewer comments, and we highlighted all the changes in the revised version of the manuscript.

The authors have addressed all the comments.

  • We thank the reviewer for his nice comment, and we hope that our manuscript gives more impact to IJMS and the readers interested in HSP proteins as therapeutic targets for cancer